# Research on Turning Motion Targets and Velocity Estimation in High Resolution Spaceborne SAR

**DOI:** 10.3390/s20082201

**Published:** 2020-04-13

**Authors:** Xuejiao Wen, Xiaolan Qiu

**Affiliations:** 1Laboratory of spatial information intelligent processing system, Institute of Electronics, Chinese Academy of Sciences, Suzhou 215000, China; xjwen@mail.ie.ac.cn; 2Institute of Electronics, Chinese Academy of Sciences, Beijing 100190, China

**Keywords:** velocity estimation, moving target, high resolution spaceborne SAR

## Abstract

The development of high resolution SAR makes the influence of moving target more prominent, which results in defocusing and other unexplained phenomena. This paper focuses on the research of imaging signatures and velocity estimation of turning motion targets. In this paper, the turning motion is regarded as the straight line motion of continuous change of moving direction. Through the analysis of the straight line motion with constant velocity and the geometric modeling of the turning motion in spaceborne SAR, the imaging signatures of the turning motion target are obtained, such as the broken line phenomenon at the curve. Furthermore, a method for estimating the turning velocity is proposed here. The radial velocity is calculated by the azimuth offset of the turning motion target and the azimuth velocity is calculated by the phase error compensated in the refocusing process. The amplitude and direction of the velocity can be obtained by using both of them. The results of simulation and GF-3 data prove the accuracy of the analysis of turning motion imaging signatures, and they also show the accuracy and validity of the velocity estimation method in this paper.

## 1. Introduction

As an active remote sensing technology, SAR has the ability to work all day and all weather, being hardly affected by bad weather or light conditions. Spaceborne SAR has been widely concerned because of its fast response, wide coverage, and strong anti-strike ability [1]. With the development of high-resolution spaceborne SAR, the influence of moving target becomes more prominent, which becomes the focus of research [2,3]. Moving targets, such as vehicles, trains, ships, and some military targets are the key targets of SAR imaging, which has very important application value in military reconnaissance, ocean observation, and environmental detection. The development of high resolution makes the influence of moving target more prominent, resulting in defocusing and other unexplained phenomena. On the one hand, it affects the interpretation and detection of the target, on the other hand, it also brings the possibility of parameter estimation of the moving target. Generally speaking, it is very important to study the moving target of high-resolution SAR.

Presently, the research on high-resolution SAR moving target mainly focuses on four aspects. The first is the analysis of imaging signatures of moving target. Most of the literatures analyzed the imaging effect of the uniform moving target, and pointed out that the target movement would lead to the target azimuth position dislocation and target defocusing [4,5,6]. There are also literature analyses of the imaging effect of moving targets with acceleration [7], that is, the third phase error and the asymmetric distortion of the image. Another part of the literature analyzes the influence of the complex motion of the target, such as the rotation motion of the target itself [8], three-dimensional motion [9,10], vibration, and other fretting forms [11]. There are also a part of literatures that introduces the influence of ship fretting, such as the ship’s three-dimensional swing, including roll, pitch, yaw, and so on [12,13]. It is pointed out that the effect of target micro motion becomes stronger with the increase of resolution, mainly due to ghosting, linear defocusing and other phenomena [14]. The second is ground moving target indication (GMTI) [15]. GMTI, as a common SAR imaging mode, has been carried on many satellite models, and it has a better performance. GMTI mainly aims at moving target detection and imaging, and the commonly used method is displaced phase center antenna (DPCA) [16], space-time adaptive processing (STAP) [17], along-track interferometry (ATI), and so on [18]. Among all kinds of moving targets, ship is just a kind of research object that attracts much attention. Ship detection is also a very popular research direction in the SAR target detection area. There are two main methods of ship detection. One is the traditional CFAR (Constant False-Alarm Rate) based detection method [19,20,21]. The other is the detection method that is based on deep learning, which has been very popular in recent years [22,23,24]. Of course, there are also attempts to combine the two to get better results [25]. The third is the estimation of moving target parameters. Most literatures have studied the velocity estimation of uniform moving target. The literature mostly concentrates on the estimation of Doppler parameters dealing with raw data [26,27,28]. For the estimation of the Doppler centroid that was determined by the radial velocity, the nominal-spectrum method, the maximum likelihood method, and the time-domain method are commonly used. For the azimuth velocity estimation, the typical approach uses the Doppler rate filters to select the velocity that better focuses the target or applies matching criteria on the displacement vector that was obtained from the spatial shift of the moving target from sub-apertures [29,30]. Some literatures also study the parameter estimation of micro moving target. Canada’s defense research and development department used autocorrelation method to estimate the rotation period of the rotating antenna target [31]. Subsequently, the Doppler frequency of vibration and rotation targets was analyzed by using smooth pseudo WVD at the University of Zurich, Switzerland. In 2011, the Air Force Academy of Engineering studied the micro Doppler effect of ground rotating parts in Bistatic SAR mode and the method of extracting micro Doppler parameters of ground vibration targets that are based on DPCA clutter suppression [32]. In recent years, the method of parameter estimation while using sparse reconstruction has also been proposed, and there are related literatures as [33,34]. The fourth and last is the focus of moving targets. Focusing imaging for moving targets has always been a hot research direction of scholars. Most of them focus on Doppler parameter estimation compensation imaging based on echo data [29]. In some literatures, compressed sensing is introduced into the focus of moving targets [35]. Some papers also study the high-resolution imaging technology of complex moving targets that are based on the imaging principle of inverse synthetic aperture radar (ISAR) [36,37]. 

Presently, the research on the moving target has been comprehensive, but there is no systematic research on the turning motion target. Turning movement is common in trains, high-speed rail, and other targets with a certain length and speed. In recent years, domestic and foreign railways have developed rapidly. In particular, China’s high-speed rail network has been built with the potential of eight horizontal and eight vertical lines, occupying the first place in the world. The scattering intensity of train, high-speed rail is large, and the signatures of its motion in SAR image are more serious with the gradual improvement of SAR resolution. The research on it will benefit the target interpretation and application of SAR image. 

This paper focuses on the research of turning motion target. By combining the geometric modeling of turning motion and the imaging geometry of spaceborne SAR, the imaging signatures of turning motion targets are analyzed. The error that is caused by turning movement is quantitatively and qualitatively analyzed. The simulation results and GF-3 images show the correctness of the imaging signatures of the turning motion target analyzed in this paper. Then a method for estimating the turning velocity is proposed. This method uses the phase error compensated in the refocusing process for azimuth velocity calculation and the azimuth offset of target out of track to estimate the radial velocity. The amplitude and direction of the velocity can be obtained by using both of them. The results of simulation and GF-3 data processing verify the correctness of the analysis and method.

The rest of this report is organized, as follows. Section 2 presents the geometry and signal models, along with the binary relationship between the velocity and residual slant range. The imaging signatures of moving target with constant velocity and turning motion target are presented here. Section 3 provides the algorithm for refocusing a moving target image and the method for estimating the velocity of the turning motion target. The results of imaging signatures of turning motion target, along with the image refocusing and velocity estimation, are presented in Section 4. Section 5 provides the conclusions.

## 2. Imaging Signatures of Turning Motion Target

### 2.1. Analysis of Moving Target with Constant Velocity in Spaceborne SAR

The instantaneous slant range varies with the azimuth time and then creates the synthetic aperture due to the relative motion between the sensor and a given target on the rotating Earth. In spaceborne SAR cases, vectors are commonly used in the calculation. Consider the geometric situation in Figure 1, which shows the satellite-Earth geometry for a given target (T) and satellite (S_1_) at a certain azimuth time. The reference coordinate system used in the figure is the inertial coordinate system. S is the satellite position when azimuth time *η*=0. The slant range history of the static target T can be expressed as:(1)$$R(\eta )=R_{s}(\eta )-R_{t}(\eta )\approx R+V_{st}\eta +\frac{1}{2}A_{st}\eta^{2}$$
where ***R_s_*(*η*)** and ***R_t_*(*η*)** denote the position vector of the satellite and the target, respectively, at a certain azimuth time. ***R***, ***V_st_***, and ***A_st_*** are, respectively, the relative range vector, velocity vector, and acceleration vector of the satellite and target at *η* = 0. Subsequently, we can obtain the value of the distance between the radar and the target, as:(2)$$\begin{array}{cc} R(\eta ) & =\left|R(\eta )\right|\\ & =R+\frac{V_{\mathrm{st}}\cdot R}{R}\eta +\frac{1}{2}\left[\frac{V_{\mathrm{st}}\cdot V_{\mathrm{st}}}{R}+\frac{A_{\mathrm{st}}\cdot R}{R}-\frac{{\left(V_{\mathrm{st}}\cdot R\right)}^{2}}{R}\right]\eta^{2}+.. \end{array}$$
where *R* is the module of the vector ***R***.

Assuming that T is moving along a straight line with constant velocity vector ***V_m_***, we can obtain the slant range of the moving target to the radar neglecting the high order items, as:(3)$$\begin{array}{cc} R^{\prime }(\eta ) & =R+\frac{\left(V_{\mathrm{st}}+V_{m}\right)\cdot R}{R}\eta \\ & +\frac{1}{2}\left\{\frac{\left(V_{\mathrm{st}}+V_{m}\right)\cdot \left(V_{\mathrm{st}}+V_{m}\right)}{R}+\frac{A_{\mathrm{st}}\cdot R}{R}-\frac{{\left[\left(V_{\mathrm{st}}+V_{m}\right)\cdot R\right]}^{2}}{R}\right\}\eta^{2}. \end{array}$$

The moving target mainly affects the history of the slant range between the target and the radar. The slant range error that is introduced by target motion is the basis for SAR moving target imaging analysis. Through the previous analysis, we have obtained the slant range history of the stationary and moving targets, respectively, in Equation (2) and Equation (3). Subsequently, the slant range error of the moving target can be calculated, as:(4)$$\begin{array}{cc} \Delta R(\eta ) & =R^{\prime }(\eta )-R(\eta )\\ & =\frac{V_{m}\cdot R}{R}\eta +\frac{1}{2}\left[\frac{2V_{\mathrm{st}}\cdot V_{m}+V_{m}\cdot V_{m}}{R}-\frac{2\left(V_{\mathrm{st}}\cdot R\right)\left(V_{m}\cdot V_{m}\right)+{\left(V_{m}\cdot R\right)}^{2}}{R^{3}}\right]\eta^{2} \end{array}$$
neglecting higher-order terms.

Decomposing ***V_m_*** orthogonally into ***V_x_*** and ***V_y_***, where ***V_y_*** directed along the radar boresight, and ***V_x_*** parallel to ***V_st_*** at azimuth time *η* = 0. We can obtain the following expressions:(5)$$\begin{array}{c} V_{x}{//V}_{\mathrm{st}};V_{x}⊥R;V_{y}⊥V_{\mathrm{st}};\\ V_{y}\cdot R=\left|V_{y}\right|\cdot R\cdot \sin\gamma \end{array}$$
where *γ* is the look angle. 

According to the relationships in Equation (5), we can rewrite Equation (4) as:(6)$$\Delta R(\eta )=\left|V_{y}\right|\sin\gamma \cdot \eta +\frac{1}{2}\left[\frac{2\left|V_{\mathrm{st}}\right|\left|V_{x}\right|+{\left|V_{x}\right|}^{2}+{\left|V_{y}\right|}^{2}\cos^{2}\gamma }{R}\right]\cdot \eta^{2}.$$

In terms of the residual range expression, the phase error and Doppler frequency error can be written as:(7)$$\begin{array}{l} \Delta \varphi \left(\eta \right)=\frac{4\pi }{\lambda }\Delta R(\eta )\\ =\frac{4\pi }{\lambda }\left\{\left|V_{y}\right|\sin\gamma \cdot \eta +\frac{1}{2}\left[\frac{2\left|V_{\mathrm{st}}\right|\left|V_{x}\right|+{\left|V_{x}\right|}^{2}+{\left|V_{y}\right|}^{2}\cos^{2}\gamma }{R}\right]\cdot \eta^{2}\right\} \end{array}$$
(8)$$\begin{array}{cc} \Delta f\left(\eta \right) & =\frac{1}{2\pi }\frac{d\Delta \varphi (\eta )}{d\eta }\\ & =\frac{2}{\lambda }\left\{\left|V_{y}\right|\sin\gamma +\left[\frac{2\left|V_{\mathrm{st}}\right|\left|V_{x}\right|+{\left|V_{x}\right|}^{2}+{\left|V_{y}\right|}^{2}\cos^{2}\gamma }{R}\right]\cdot \eta \right\}. \end{array}$$

By analyzing the Doppler frequency error in Equation (8), we can get the imaging signatures of moving target SAR image, which are azimuth delocalization and azimuth defocusing phenomenon described in detail in the following paragraph. 

#### 2.1.1. Target Azimuth Delocalization

From the Doppler frequency error expression in Equation (8), we can obtain the Doppler centroid frequency error, as:(9)$$\Delta f_{dc}=\frac{2\sin\gamma }{\lambda }\left|V_{y}\right|.$$

According to the SAR imaging theory, the azimuth position of the target is related to the Doppler centroid frequency The Doppler centroid frequency error that was caused by the moving target will result in azimuth delocalization, which is related to Doppler modulation frequency and beam scanning speed, namely ground speed.

Let *f_r_* denotes the Doppler modulation frequency and *V_g_* denotes the ground speed. The azimuth position offset is shown, as:(10)$$\Delta x=\frac{\Delta f_{dc}}{f_{r}}V_{g}.$$

It can be seen from Equation (10) and Equation (9) that the factors affecting the azimuth position offset include the target radial velocity, the wavelength, the ground beam scanning speed, the Doppler modulation frequency, and the look angle. This kind of azimuth delocalization of the target is usually called location error. 

Many factors affect the location error of the target, among which the radial velocity of the target relative to the radar platform is an important aspect. It is the radial velocity of the target that causes the change of Doppler centroid frequency. The Doppler centroid frequency error is reflected as target azimuth delocalization in the SAR image. There is an approximately linear relationship between the target’s azimuth position offset and radial velocity.

#### 2.2.2. Target Azimuth Defocusing

From the Doppler frequency error expression in Equation (11), we can get the Doppler modulation frequency error, as:(11)$$\Delta f_{r}=\frac{2}{\lambda R}\left(2\left|V_{\mathrm{st}}\right|\left|V_{x}\right|+{\left|V_{x}\right|}^{2}+{\left|V_{y}\right|}^{2}\cos^{2}\gamma \right).$$

A matched filter mainly realizes the SAR azimuth imaging process. The design of matched filter is related to the Doppler information of signal, and the Doppler frequency modulation information is the main one. The target motion causes the change of the Doppler frequency modulation. The azimuth processing of the echo signal of the moving target with the Doppler frequency modulation rate of the stationary target will make it difficult to obtain the focused image due to the mismatch of the matching filter. Equation (11) shows that the most important factor affecting the Doppler frequency modulation is the azimuth velocity. Apparently, the azimuth velocity will generate a loss of focus in SAR image.

Through the above analysis, we can conclude that the signatures of moving target with constant velocity in SAR image are defocusing phenomenon mainly caused by azimuth velocity and azimuth delocalization mainly caused by radial velocity.

### 2.2. Analysis of Turning Motion Target

In this section, the turning motion model of multi-point targets, which is derived from the turning motion of train, will be constructed and its imaging signatures will be analyzed. 

The geometric model of turning motion target is built in the ground plane in order to facilitate imaging analysis, which is shown in Figure 2. For the convenience of subsequent expression, the plane rectangular coordinate system is established in the ground plane, with the center point O of the scene as the origin of the coordinate system, the distance to the ground as the x-axis, and the azimuth position as the y-axis.

Generally, the curve is mainly composed of three parts. They are the straight part before entering the curve, the arc part of the curve, and the straight part leaving the curve. The two straight line parts are tangent to the arc part at the connection point. According to the geometry knowledge of a curve, four factors that determine the specific shape of a curve are:centering angle of arc part of curve, mark as *β*;radius of the arc part of the curve, mark as *R_l_*;slope of line l1 at the beginning of the curve, mark as θ1; and,slope of line l2 at the end of the curve, mark as θ2.

Section AB is the straight track before entering the curve, OC is the straight track after leaving the curve, and BO is the curve between the two straight tracks, as shown in Figure 2. According to the curve geometry knowledge, the rotation center of the curve is the O’ point marked in Figure 2, which is the intersection of the vertical line *l_1_* of the straight line AB and the vertical line *l_2_* of the straight line OC. O’B and O’O are the vector *R_l_* of the curve, and the angle *β* is the arc center angle corresponding to the arc BO. 

In fact, the part of the arc can be regarded as a small line segment with continuous change of slope. Subsequently, the turning movement is actually a "straight line movement" with continuous change of moving direction. Equation (6) shows that the slant range history error is mainly determined by the azimuth velocity component and the radial velocity component of the moving target. Taking a scattering point in any part of the curve as an example, where the target position is (*r_t_,a_t_*) and the corresponding tangent slope is *θ_t_*, then we can get:(12)$$\begin{array}{l} a_{t}=a_{0}-R_{l}\cos\theta_{t}\\ r_{t}=r_{0}-R_{l}\sin\theta_{t}. \end{array}$$

In the process of uniform turning movement, the component of velocity in azimuth and radial direction is changing due to the change of movement direction. The slant range history error relative to radar platform can be deduced from the range history error of linear moving target in Equation (6), as:(13)$$\Delta R_{t}(\eta )\approx V_{y}(\eta )\sin\gamma \cdot \eta +\frac{1}{2}\left[\frac{2\left|V_{st}\right|V_{x}(\eta )+V_{x}^{2}(\eta )+V_{y}^{2}(\eta )\cos^{2}\gamma }{R}\right]\cdot \eta^{2}$$
where *V_y_*(*η*) and *V_x_*(*η*) represent the azimuth velocity component and radial velocity component, respectively, which are determined by the direction of motion:(14)$$\begin{array}{l} V_{y}(\eta )=V_{m}\cdot \cos\left[\vartheta \left(\eta \right)\right]\\ V_{x}(\eta )=V_{m}\cdot \sin\left[\vartheta \left(\eta \right)\right] \end{array}$$
where *V_m_* is the amplitude of target turning velocity and *ϑ*(*η*) is the target turning direction. In the curve part, the instantaneous motion direction of the target is the tangent direction of the target position. It is assumed that the target direction of motion at *η*=0 is *θ_0_*. According to the curve arc length and the amplitude of target velocity, the instantaneous direction of the target’s motion can be calculated, as:(15)J(η)=θ0+θ2−θ1αRlVmη.

Take Equation (15) into Equation (14), and rewrite Equation (13) as:(16)ΔRt(η)=Vm⋅cos[θ0+θ2−θ1αRlVmη]sinγ⋅η+12[2|Vst|Vm⋅sin[θ0+θ2−θ1αRlVmη]+Vm2sin2[θ0+θ2−θ1αRlVmη]+Vm2cos2[θ0+θ2−θ1αRlVmη]cos2γR].

The sine function and cosine function in Equation (16) are expanded by Taylor at *η*=0 to obtain:(17)$$\begin{array}{cc} \cos\left(\theta_{0}+\frac{\theta_{2}-\theta_{1}}{\alpha R_{l}/V_{m}}\eta \right) & =\cos\theta_{0}-\left(\frac{\theta_{2}-\theta_{1}}{\alpha R_{l}/V_{m}}\sin\theta_{0}\right)\eta \\ & +\left[\frac{1}{2}{\left(\frac{\theta_{2}-\theta_{1}}{\alpha R_{l}/V_{m}}\right)}^{2}\cos\theta_{0}\right]\eta^{2}+\ldots \\ \sin\left(\theta_{0}+\frac{\theta_{2}-\theta_{1}}{\alpha R_{l}/V_{m}}\eta \right) & =\sin\theta_{0}+\left(\frac{\theta_{2}-\theta_{1}}{\alpha R_{l}/V_{m}}\cos\theta_{0}\right)\eta \\ & -\left[\frac{1}{2}{\left(\frac{\theta_{2}-\theta_{1}}{\alpha R_{l}/V_{m}}\right)}^{2}\sin\theta_{0}\right]\eta^{2}+\ldots . \end{array}$$

The polynomial representation of Equation (16) can be obtained by taking Equation (17) into the Equation (16):(18)$$\Delta R_{t}(\eta )=p_{0}+\cos\theta_{0}\cdot V_{m}\sin\gamma \cdot \eta +p_{2}\eta^{2}+\ldots ,$$

This calculation process is relatively tedious, therefore only the specific expression of the coefficient of the primary term is given here. The coefficients of other secondary terms are replaced by *p_n_*.

In terms of the residual range expression, the phase error, and Doppler frequency error can be written as:(19)$$\Delta \varphi \left(\eta \right)=\frac{4\pi }{\lambda }(p_{0}+\cos\theta_{0}\cdot V_{m}\sin\gamma \cdot \eta +p_{2}\eta^{2})+\ldots$$
(20)$$\Delta f\left(\eta \right)=\frac{2}{\lambda }(\cos\theta_{0}\cdot V_{m}\sin\gamma +2p_{2}\eta )+\ldots .$$

The errors in polynomial form are mainly the target azimuth position offset caused by primary term and the target azimuth defocusing phenomenon caused by higher term. Azimuth defocusing is a qualitative description, which can not be clearly matched in the results of multi-point target motion imaging. Therefore, this section focuses on the quantitative analysis of multi-point target azimuth location error.

According to Equation (10) and Equation (20), the azimuth position offset that is caused by turning movement of point target with position (*r_t_*,*a_t_*) can be obtained, as follows:(21)$$\Delta a=\frac{\Delta f_{dc}}{f_{r}}V_{g}=\frac{2/\lambda \cdot \cos\theta_{0}\cdot V_{m}\sin\gamma }{f_{r}/V_{g}}.$$

It can be seen from Equation (21) that the azimuth position offset of the turning moving target is related to the slope of the tangent corresponding to the position at the time of azimuth center. *θ_0_* is a variable that changes with the target point, which can be expressed as:(22)$$\theta_{0}=\theta \left(r^{\prime },a^{\prime }\right)$$
where (*r*’,*a*’) represents the stationary position of the target at any point of the turning movement. Therefore, the azimuth position offset of certain target can be obtained as:(23)$$\Delta a(r^{\prime },a^{\prime })=\frac{2/\lambda \cdot \cos\theta \left(r^{\prime },a^{\prime }\right)\cdot V_{m}\sin\gamma }{f_{r}/V_{g}}.$$

In fact, the product of *V_m_* and direction function cosθ(*r*’,*a*’) is the radial velocity of each target on the curve. The amplitude of the whole velocity *V_m_* is constant, so the different direction θ(*r*’,*a*’) of each target will lead to different radial velocity, which results in a different azimuth position offset. This different offset will make the targets no longer hold the shape of the curve, but the broken line of different shapes which depends on the magnitude of the velocity.

The specific imaging effects will be detailed in the following simulation results in Section 4.1.

## 3. Moving Target Refocusing and Velocity Estimation of Turning Motion Target

Figure 2 shows the turning motion model in this paper. In this paper, the turning motion is divided into three parts, two of which are uniform linear motion, and one is uniform turning motion. The speed of turning is equal to that of straight-line motion because the targets of turning motion are a whole (such as a train). Based on this assumption, we can determine the speed of turning by estimating the speed of a certain straight-line movement.

The velocity of linear moving target can be divided into radial velocity and azimuth velocity. This paper deals with formed SLC SAR slices, that is, the region of interest (ROI). There are many methods of velocity estimation for this kind of moving target, among which the azimuthal velocity estimation is the most popular, while the radial velocity estimation is less studied and it has its application premise. In this paper, according to the signatures of the turning motion target, known is the original position of the target (for example, the rail in the image is the position where the train should be when it is stationary), the radial velocity of the target is calculated by the known target azimuth position offset. At the same time, the motion compensation algorithm that is based on autofocusing is used to refocus ROI target. The azimuth velocity of the target is calculated by the motion compensation phase and radial velocity of the target, combined with Equation (7). Figure 3 shows the overall algorithm flow chart. We number each computational block in order to distinguish the calculation steps of the method and expand the discussion of the method in detail. The italics in the Figure 3 represent the data dimension and domain output after the computational block, which is also the input of the next computational block. The process is briefly described, as follows:

1. intercept the target of the linear motion part as the ROI target, which size is M × N, M refers to azimuth direction while N refers to range direction;

2. obtain the equivalent raw data in Doppler domain sized M × N by applying inverse azimuth Fourier transform (IFFT) to ROI target;

3. autofocusing for the data obtained in the previous step through the minimum entropy phase adjustment algorithm and obtain the compensation phase curve in Doppler domain;

4. Obtain the refocused ROI target sized M × N in time domain by applying azimuth Fourier transform (FFT);

5. measure and calculate the displacement of azimuth position caused by target motion;

6. calculate the radial velocity *V_y_* according to Equation (9) and Equation;

7. The azimuth velocity *V_x_* is calculated by the compensation phase curve and the radial velocity estimation results combined with Equation (7); and,

8. calculate the amplitude *V_m_* and the direction *θ_m_* of the velocity through the radial velocity *V_y_* and the azimuthal velocity *V_x_*.

The motion compensation algorithm and velocity estimation algorithm based on autofocusing are described in detail below.

### 3.1. Motion Compensation Algorithm

This section, corresponding to steps 1–4 in Figure 3, is aimed to obtain the refocused moving target by autofocusing method, and to obtain the compensation phase curve that is caused by motion error, which is used to estimate the velocity of the target. The following are steps in detail.

Step 1. Obtain the ROI target

We have a SAR complex image of a turning motion target. We intercept the linear motion part from the formed image as ROI target. Subsequently, the following technique used in the paper deals with the formed SAR ROI target. The size of the ROI target is M × N, M refers to azimuth direction, while N refers to range direction, and the initial state is in time domain.

Step 2. Azimuth IFFT

The ROI target is inverted in order to obtain the equivalent raw data, which are the inputs of the motion compensation technique that were acquired by applying inverse Fourier transform (IFFT) and inverse compression in the azimuth direction. Thus, we can obtain the data in the Doppler frequency domain before the azimuth compression, as:(24)$$s(t,f_{\eta })=\sigma (f_{\eta })\cdot \sin c\left\{\pi B\left[t-\frac{2R(f_{\eta })}{c}\right]\right\}\cdot \exp\left[\frac{-j4\pi R(f_{\eta })}{\lambda }\right]$$
where *f_η_* is the Doppler frequency, *t* is the range time, *σ* is the backscattering coefficient, *λ* is the radar wavelength, *c* is the light speed, and *B* is the bandwidth of the transmitted signal.

The target is assumed to be a point-like scatterer, and the motion errors that are caused by the platform are assumed to have been perfectly compensated by chirp-scaling algorithm. Thus, we obtain the range migration, as:(25)$$R\left(f_{\eta }\right)=R+\Delta R(f_{\eta })$$
where *R* is the slant range at synthetic aperture time *η*=0, as mentioned in Section 2. Δ*R*(*f_η_*) is the residual slant range expression that is caused only by the moving target in the Doppler domain.

Step 3. The minimum entropy phase adjustment autofocusing algorithm

The range migration causes a variation in the phase terms. For the compensation of the time-varying Doppler phase, the minimum entropy phase adjustment based on the gradient descent algorithm is chosen [38]. This is found by iteratively solving an equation, which is derived by minimizing the entropy of the image and ultimately provides the compensating phase curve. It is worth mentioning that the compensated phase error curve is obtained in the azimuth Doppler domain, whose dimension is M × 1, that is, it changes along the azimuth direction.

Step 4. Azimuth FFT

The data after motion compensation can be written, as:(26)$$s(t,f_{\eta })=\sigma (f_{\eta })\cdot \sin c\left\{\pi B\left[t-\frac{2R}{c}\right]\right\}\cdot \exp\left(\frac{-j4\pi R}{\lambda }\right).$$

Finally, for the image formation, an FFT along the azimuth direction can be used to refocus the image.

### 3.2. Velocity Estimation Method

This section deals with the estimation velocity method that corresponds to steps 5–8 in Figure 3. The velocity estimation method in this section is based on the error analysis presented in Section 2 and the motion compensation technique in the previous section. The followings are steps in detail.

Step 5. Measure the displacement of azimuth position

We have known the correct position of the target when it is stationary (for example, the rail is the position where the train should be when it is stationary). By comparing the position of the moving target with the position it should be when it is stationary along the azimuth direction, we can obtain the number of azimuth offset points as ΔB. It is assumed that the azimuth spatial resolution of the image is *ρ_a_*, and we can get the azimuth offset expressed in meters, as:(27)$$\Delta a=\Delta B\cdot \rho_{a}.$$

Step 6. Calculate the radial velocity Vy

The radial velocity of the target is calculated according to the offset of the turning moving target from the original position, which is the target azimuth position offset Δ*a*. We can obtain the *V_y_* according to Equation (9) and Equation (10):(28)$$V_{y}=\frac{\Delta a\lambda f_{r}}{2V_{g}\sin\gamma }.$$

Step 7. Calculate the azimuth velocity Vx

The basic principle of the velocity estimation method in this step is that, when the defocused ROI target is processed by autofocusing, the imaging quality is obviously improved and the focusing effect is good, the phase error caused by motion are completely compensated, that is:
(29)$$\Delta \varphi (f_{\eta })\approx \Delta \varphi (m)$$
where Δ*φ*(*f_η_*) represents the phase error of azimuth frequency domain that is caused by motion. Δ*φ*(*m*) represents the compensated phase obtained by the phase correction algorithm. 

Δ*φ*(*f_η_*) contains parameters related to the velocity of motion. As long as the specific analytical expression of Δ*φ*(*f_η_*) is obtained, the corresponding velocity estimates can be obtained.

The analytical expression for Δ*φ*(*f_η_*) is derived below. In order to facilitate the analysis, the two-dimensional spectrum of the state target echo is rewritten, as follows:(30)$$ss(R,f_{\eta })=G_{rcm}(f_{\eta },R)\cdot G_{ac}(f_{\eta },R)\cdot G_{o}(f_{\eta },f_{\tau },R)$$
where *G_rcm_*(*f_η_*,*R*) is the range migration term in azimuth frequency domain. *G_ac_*(*f_η_*,*R*) is the azimuth modulation term. *G_o_*(*f_η_*,*R*) is the second distance compression term. The expression of the three is:(31)$$\begin{array}{l} G_{rcm}(f_{\eta },R)=\frac{j4\pi R}{cD(f_{\eta },V_{r})}f_{\tau }\\ G_{ac}(f_{\eta },R)=\frac{j4\pi R}{\lambda }D(f_{\eta },V_{r})\\ G_{o}(f_{\eta },f_{\tau },R)=\frac{j4\pi R}{c}\cdot \frac{f_{\tau }^{2}}{2f_{0}D^{3}(f_{\eta },V_{r})}\cdot \frac{c^{2}f_{\eta }^{2}}{4V_{r}^{2}f_{0}}\\ D(f_{\eta },V_{r})=\sqrt{1-\frac{\lambda^{2}f_{\eta }^{2}}{4V_{r}^{2}}} \end{array}$$
where *f_τ_* is the range frequency, *V_r_* is the equivalent velocity used in the imaging algorithm, and *f_0_* is the carrier frequency.

The Doppler frequency error of moving target obtained from Section 2 is:(32)$$\begin{array}{l} \Delta f_{dc}=\frac{2\sin\gamma }{\lambda }\left|V_{y}\right|\\ \Delta f_{r}=\frac{2}{\lambda R}\left(2\left|V_{st}\right|\left|V_{x}\right|+{\left|V_{x}\right|}^{2}+{\left|V_{y}\right|}^{2}\cos^{2}\gamma \right). \end{array}$$

In the basic theory of SAR imaging, the Doppler frequency information of the target represents the relationship between the azimuth time and the azimuth frequency of the target point. It can be considered that the azimuth frequency of moving target has changed as compared with that of stationary target, thus changing the expression of target echo in two-dimensional frequency domain. If the azimuth frequency of a moving target is *f_η_*’ and that of a stationary target is *f_η_*, the relationship between them can be approximately considered as:(33)$$f_{\eta }{}^{\prime }=(1+\Delta f_{dc}/f_{R})f_{\eta }+\Delta f_{dc}.$$

The azimuth modulation term of moving target in azimuth frequency domain are:(34)$$G_{ac}{}^{\prime }(f_{\eta },R)=\frac{j4\pi R}{\lambda }D((1+\Delta f_{dc}/f_{R})f_{\eta }+\Delta f_{dc},V_{r}).$$

After the matched filtering of the imaging algorithm, the residual phase error is:(35)$$\Delta \varphi (f_{\eta },R)=G_{ac}{}^{\prime }(f_{\eta },R)\cdot G_{ac}{}^{*}(f_{\eta },R)$$
where *G^*^_ac_*(*f_η_*,*R*) is the matched filter in the frequency domain of the azimuth of the stationary target. The results of Equation (35) are expanded by Taylor series on *f_η_* = 0 and retained to the quadratic term, as:(36)$$\Delta \varphi (f_{\eta },R)=-\frac{j\pi R^{2}\lambda }{2V_{r}{}^{2}}\left[2\Delta f_{dc}\left(1+\frac{\Delta f_{r}}{f_{R}}\right)f_{\eta }+\left(\frac{2\Delta f_{r}}{f_{R}}+\frac{\Delta f_{r}{}^{2}}{f_{R}{}^{2}}\right)f_{\eta }{}^{2}\right].$$

The linear part of the phase error compensation term is not accurate due to the motion compensation algorithm, so only the quadratic term of the phase error compensation term can be used for velocity estimation. 

Using *f_η_* as the independent variable to do quadratic polynomial fitting to Δ*φ*(*m*) to obtain the coefficient of quadratic term *p*_2_:
(37)$$p_{2}\approx -\frac{2\pi R}{V_{r}{}^{2}f_{R}}\left(2\left|V_{st}\right|V_{x}+V_{x}{}^{2}+V_{y}{}^{2}\cos^{2}\gamma \right).$$

The parameters in the Equation (37) are all known, and the result of *V_x_* can be obtained by solving the quadratic equation of one variable.

Step 8. Calculate the amplitude and direction of the estimated velocity

The amplitude and direction of velocity are obtained, as follows, using *V_y_* and *V_x_*:(38)$$\begin{array}{l} V_{m}=\sqrt{V_{x}^{2}+V_{y}^{2}}\\ \theta_{m}=\arctan\left(\frac{V_{x}}{V_{y}}\right) \end{array}$$
where *V_m_* and *θ_m_* denote the amplitude and direction of the velocity, respectively.

## 4. Results

### 4.1. Simulation Results and Actual Data of Imaging Signatures of Turning Motion

This section mainly shows the simulation results of turning movement at different turning speeds to prove the accuracy and rationality of the analysis in Section 2. Figure 2 shows the shape of simulation curve. Table 1 shows the simulation curve parameters. Table 2 shows the simulation radar parameters. In this simulation experiment, a total of twenty-three point targets are simulated, including seven straight lines entering the curve, nine curves, and seven straight lines out of the curve.

Figure 4a shows the results of static curve target. It can be seen that the turning model is relatively accurate. In this experiment, three turning velocities are simulated, which are 20 km/h, 45 km/h, and 65 km/h. Figure 4b–d show the imaging results of the three turning velocities. The static curve targets and the moving targets are marked in the figure in order to clearly see the difference between the turning moving targets and the original static curve targets. 

It can be seen from the Figure 4b–d that the three segments of the turning movement show two different forms. The linear part of the in and out curves show the defocusing and the same azimuth position offset. The curve part shows different azimuth offset with the change of target position. It is this kind of continuous and inconsistent delocalization that leads to the phenomenon of broken line, so that the curved part cannot be smoothly connected with the straight part.

This kind of different azimuth offset makes the moving target of the curve show a line shape, rather than the original curved shape. When compared with the results of different velocities, it can be seen that the larger the amplitude of velocity is, the more obvious the deflection trace of the curve target is, and the more inconsistent the shape of the original curve is. When the amplitude of velocity reaches a certain value (45 km/h in the simulation), the moving target of the curve can even appear the phenomenon of approximately right angle bend. The broken line phenomenon proves that the analysis in Section 2 is correct.

In the 3m resolution image of GF-3, we also found that the turning movement, which caused the broken line phenomenon, as shown in Figure 5. It can be seen from the optical image that the red part in the Figure 5b is a railway track, so the brighter line in the Figure 5a is supposed to be a turning train. 

It can be seen from the figure that the train is separated from the rail track. The track is a normal curve model with two straight sections connecting with the curve section smoothly, while the train shows three broken lines. The straight parts of both ends of the train are consistent with the straight parts of the rail, and there is an identical azimuth position offset, which is the signatures of the uniform linear moving targets. However, the curve section of the train is completely different from the original rail track shape, which is caused by the continuous change of the radial velocity during the turning, resulting in the continuous change of the azimuth position offset. Different position targets have a different azimuth position offset, which will make the targets no longer hold the shape of the curve, but the broken line of different shapes, which depends on the magnitude of the velocity. Therefore, the imaging train’s shape is different from the original shape, showing a broken line shape rather than an arc shape.

Through the analysis of Section 2.1 and Section 2.2, it can be known that the radial velocity determines the target azimuth position offset, while the radial velocity of the turning moving target is related to the amplitude of the target velocity and the moving direction. The reason why the train in Figure 5 can have a broken line phenomenon of approximately right angle is that the continuous change of azimuth position offset that is caused by its moving velocity and direction is just matched, as good as the result of simulation v = 50km/h in Figure 4c. If the speed increases or decreases, it will also show the same phenomenon in the simulation, as in Figure 4b,d.

Generally speaking, the phenomenon of train broken line in the Figure 5 is caused by different azimuth position offset when turning. This is consistent with the simulation results. It shows that the analysis of imaging signatures of turning motion in this paper is correct.

### 4.2. Simulation Results and Actual Data Results of Velocity Estimation

This section mainly shows the velocity estimation results of turning movement to prove the accuracy and effectiveness of the method presented in Section 3. The experimental data are the moving target data of three turning velocities generated by the simulation in the previous section. In this experiment, the straight-line part of the entrance curve is used for velocity estimation.

We intercept the straight part of the turning motion for autofocusing, which has been circled in red in the Figure 4b–d. For convenience, only the result of V = 45km/ h is shown here as an example. Figure 6a–b shows the comparison before and after the focus of the moving target. It can be seen that the image quality of the point target after the autofocus has improved significantly. We extract this target for quality evaluation. The quality evaluation results are shown in Table 3 and the range and azimuth waveforms are shown in Figure 6c–d. It can be seen that the azimuth waveform and the range waveform meet the index, indicating that the phase error that is caused by the motion has been compensated. The compensation phase is shown in Figure 6e. At the same time, we measure the azimuth position offset between the moving point target and the original stationary target. Using the Equation (28) and Equation (37), we can calculate the velocity estimation result. Table 4 shows the velocity estimation results of simulation. The estimation results are basically consistent with the actual simulation velocity, which proves the correctness of the proposed method. 

The following is a brief analysis and assessment of the simulation results. 

Firstly, the target motion compensation algorithm that is based on autofocusing is exact and effective. The point target result is evaluated to be in accord with the index, as shown in Table 3 and Figure 6, which illustrates that the algorithm performs well. Next, the velocity estimation method is effective. Although there are small errors between the velocity estimation results and the real target velocity and direction, they are generally consistent with each other, which illustrates that the velocity estimation method performs well. Finally, we discuss the reason of the velocity estimation error. It can be seen from the results that the error of velocity estimation is 0.15–0.25 m/s (we converted the unit km/h in Table 4 into m/s) and 0.5–3 degrees. In fact, the error mainly comes from two ways, which are the error caused by the azimuth and radial velocity estimation method. On the one hand, the azimuth position offset is obtained by measuring the offset points and azimuth resolution, and its accuracy is limited by the resolution. The error offset of 3m calculated from simulation parameters is enough to produce the error of 0.05–0.1m/s; on the other hand, the azimuth velocity is calculated by the quadratic term of the phase error compensation curve fitting. There are small errors in fitting process and approximate omissions in equation calculation, which leads to the estimation error of azimuth velocity. We will continue to explore in-depth in the future research.

The actual data still select the train target in Figure 5a. Table 5 shows the parameters of the data. The straight-line part before entering the curve is autofocused, and the results are shown in Figure 7. It can be seen that the quality of train target has been improved after autofocusing as the enlarged part shown by the red line in Figure 7a–b. Figure 7c shows the phase error curve. We measured the displacement of the train and the rail track along the azimuth direction that is shown in Figure 7d. Using the Equation (28) and Equation (37), we can obtain the velocity estimation results that are shown in Table 6.

The rail track is located near Guanting Reservoir in Beijing with longitude E115.61 and latitude N40.22, which can be found in the optical map. The direction of the track is approximately 40–50 degrees west north, which is not much different from our estimation result. The estimated amplitude of velocity of the train is 65.7 km/h, which is a reasonable value for the freight train. It can show the correctness and validity of the method.

## 5. Conclusions

This paper mainly studies the imaging signatures of SAR image of turning motion, and it points out that the turning motion will produce the phenomenon of broken line. In this paper, the turning velocity is estimated by using the displacement of the turning movement, which deviates from the original position and the phase error obtained by autofocusing. The results of the simulation and actual data prove the accuracy of the analysis of turning motion imaging signatures, and also show the accuracy and validity of the velocity estimation method in this paper.

## Figures and Tables

**Figure 1 sensors-20-02201-f001:**
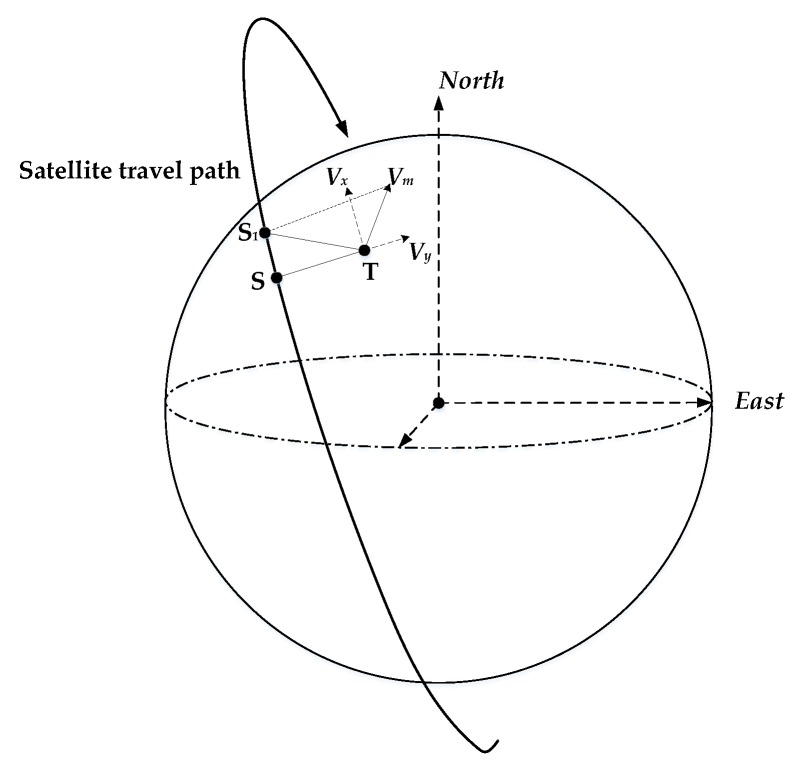
Satellite-Earth geometry at certain time.

**Figure 2 sensors-20-02201-f002:**
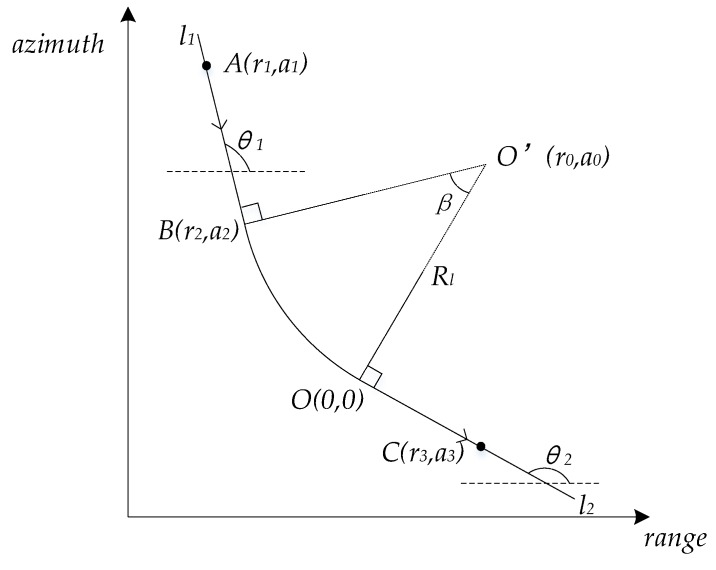
Geometric model of turning motion targets.

**Figure 3 sensors-20-02201-f003:**
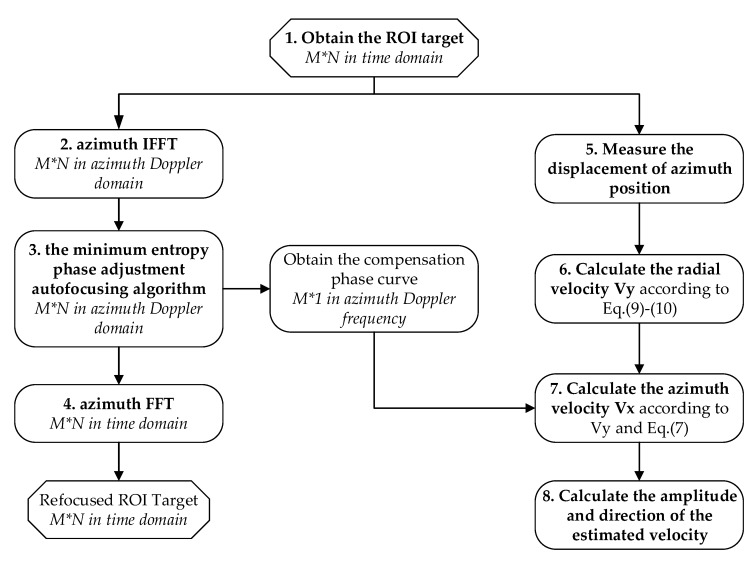
The flowchart of velocity estimation and target refocusing.

**Figure 4 sensors-20-02201-f004:**
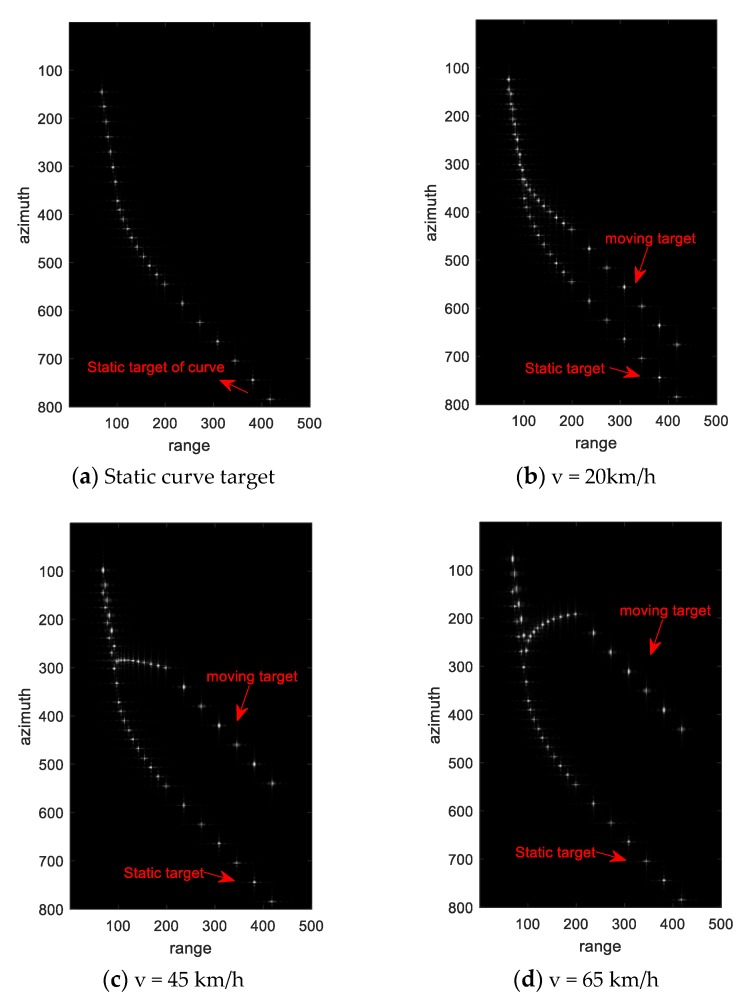
Simulation results of turning motion targets with velocity 20 km/h, 45 km/h and 65 km/h.

**Figure 5 sensors-20-02201-f005:**
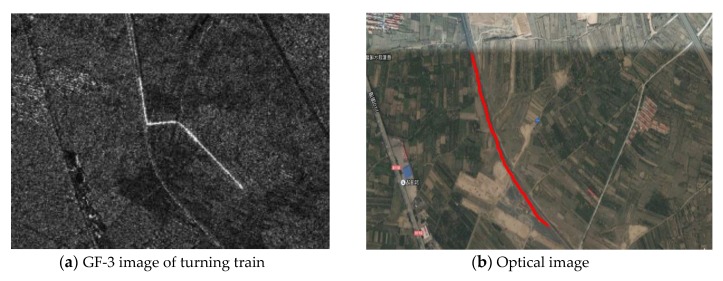
GF-3 SAR image and its corresponding optical image.

**Figure 6 sensors-20-02201-f006:**
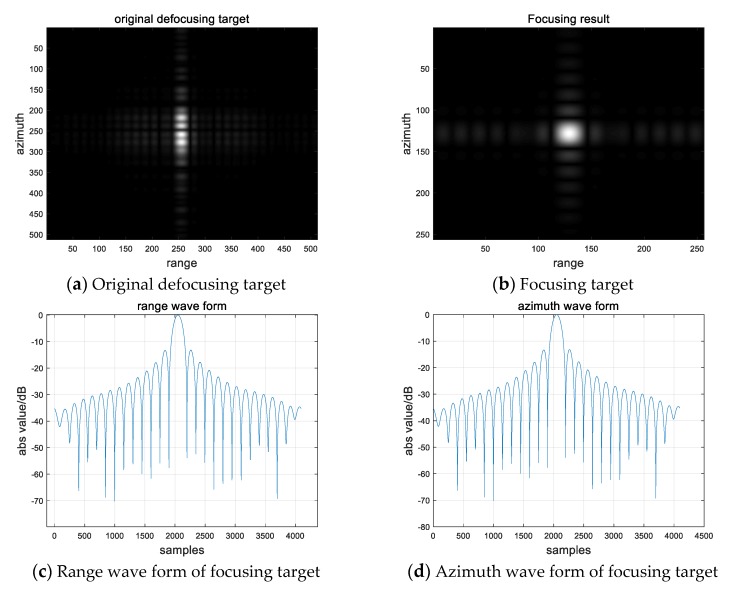
Processing result of moving target.

**Figure 7 sensors-20-02201-f007:**
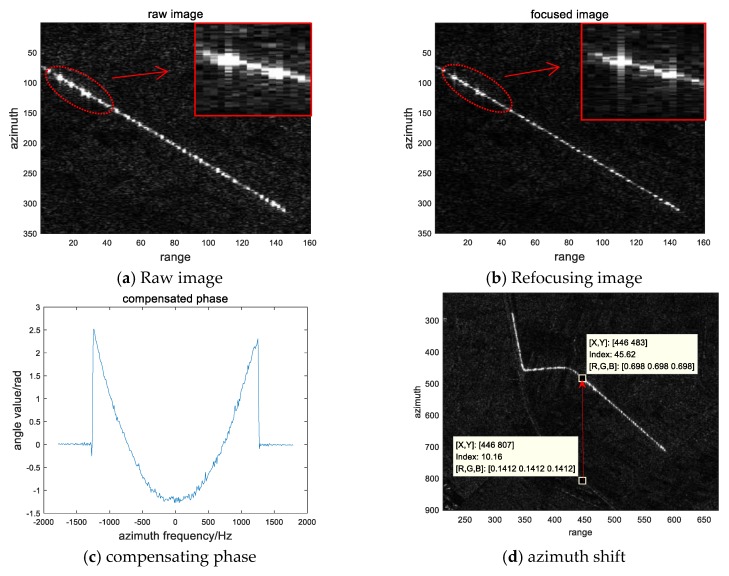
Processing of the moving train data.

**Table 1 sensors-20-02201-t001:** Simulation curve parameters.

*R_l_*	*β*	*θ* _1_	*θ* _2_
988.6375 m	30.3°	−82.8°	−52.5°

**Table 2 sensors-20-02201-t002:** Simulation radar parameters.

Parameters	Value	parameters	value
λ	0.03125	PRF	3205.128
Transmit Band	110 MHz	Lookangle	42.41°
Sample Rate	120 MHz	Vst	7368.9706 m/s
Reference Range	577368.962661	Vg	7046.7001 m/s
Aperture Time	0.538357 s	fr	4845.9846
*ρ_a_*	3 m	*ρ_r_*	1.25 m

**Table 3 sensors-20-02201-t003:** Quality evaluation of autofocusing target.

Dimension	Resolution	PLSR	ILSR
range	1.182 m	−13.882 dB	−9.998 dB
azimuth	2.212 m	−13.052 dB	−10.035 dB

**Table 4 sensors-20-02201-t004:** Velocity estimation results.

V	Pos Shift	Vy	Vx	Arctan(Vx/Vy)	Vm
V = 20 km/h	217.6585 m	3.4653 m/s	4.6258 m/s	53.1618°	20.807 km/h
V = 45 km/h	505.6712 m	8.0508 m/s	9.7932 m/s	50.5769°	45.639 km/h
V = 65 km/h	743.1169 m	11.8312 m/s	13.8527 m/s	49.5000°	65.582 km/h

**Table 5 sensors-20-02201-t005:** GF-3 radar parameters.

Parameters	Value	Parameters	Value
λ	0.055517	PRF	3566.333984
Transmit Band	82.5 MHz	Lookangle	57.61°
Sample Rate	90 MHz	Vst	7662.5171 m/s
Reference Range	651926.425839	Vg	7094.4612 m/s
Aperture Time	0.4684 s	fr	5337.0810

**Table 6 sensors-20-02201-t006:** Processing results.

Items	Estimate Results
azimuth shift	499.9642 m
V_y_	12.3641 m/s
V_x_	14.2399 m/s
Arctan(V_x_/V_y_)	49.0331°
V_m_(m/s)	18.8586 m/s
V_m_(km/h)	67.891 km/h

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
