# Peer review of "Research on Turning Motion Targets and Velocity Estimation in High Resolution Spaceborne SAR"

_sensors, 2020, doi:10.3390/s20082201_

Round 1

Reviewer 1 Report

The paper develops a new method to focus on the turning motion targets and estimate its velocity using high resolution spaceborne SAR. The application of high resolution SAR in this field is interesting. I have several minor points of the manuscript: 1) Figure 1 was not clear, and other figures should also make improvement (especially words in the figure). 2) Is it possible to distinguish of different moving targets regarding the spatial resolution of the SAR? Please discuss it. 3) The symbols mentioned in line 169-172 were not show in figure 2. 4) Even if the estimated results were right, however, there were still some discrepancy with simulation results. Please include discussion or comment on it.

Reviewer 2 Report

See attached file

Round 2

Reviewer 1 Report

The paper has been well improved and can be accepted for publication.